# Efficacy and Toxicity of Different Chemotherapy Protocols for Concurrent Chemoradiation in Non-Small Cell Lung Cancer—A Secondary Analysis of the PET Plan Trial

**DOI:** 10.3390/cancers12113359

**Published:** 2020-11-13

**Authors:** Eleni Gkika, Stefan Lenz, Tanja Schimek-Jasch, Cornelius F. Waller, Stephanie Kremp, Andrea Schaefer-Schuler, Michael Mix, Andreas Küsters, Marco Tosch, Thomas Hehr, Susanne Martina Eschmann, Yves-Pierre Bultel, Peter Hass, Jochen Fleckenstein, Alexander Henry Thieme, Marcus Stockinger, Karin Dieckmann, Matthias Miederer, Gabriele Holl, Hans Christian Rischke, Sonja Adebahr, Jochem König, Harald Binder, Anca-Ligia Grosu, Ursula Nestle

**Affiliations:** 1Department of Radiation Oncology, Medical Center—University of Freiburg, Robert-Koch-Str. 3, 79106 Freiburg, Germany; tanja.schimek-jasch@uniklinik-freiburg.de (T.S.-J.); christian.rischke@uniklinik-freiburg.de (H.C.R.); sonja.adebahr@uniklinik-freiburg.de (S.A.); anca.grosu@uniklinik-freiburg.de (A.-L.G.); Ursula.Nestle@mariahilf.de (U.N.); 2German Cancer Consortium (DKTK) Partner Site Freiburg, German Cancer Research Center (DKFZ), Neuenheimer Feld 280, 69120 Heidelberg, Germany; 3Faculty of Medicine, University of Freiburg, 79106 Freiburg, Germany; cornelius.waller@uniklinik-freiburg.de (C.F.W.); Michael.mix@uniklinik-freiburg.de (M.M.); 4Institute of Medical Biometry and Statistics, Faculty of Medicine and Medical Center, University of Freiburg, 79106 Freiburg, Germany; lenz@imbi.uni-freiburg.de (S.L.); binderh@imbi.uni-freiburg.de (H.B.); 5Department of Medicine I, Hematology, Oncology and Stem Cell Transplantation, Medical Center-University of Freiburg, 79106 Freiburg, Germany; 6Department of Radiotherapy and Radiation Oncology, Saarland University Medical Center and Faculty of Medicine, 66421 Homburg/Saar, Germany; Stephanie.Kremp@uks.eu (S.K.); Jochen.Fleckenstein@uks.eu (J.F.); 7Department of Nuclear Medicine, Saarland University Medical Center and Faculty of Medicine, 66421 Homburg/Saar, Germany; andrea.schaefer@uks.eu; 8Department of Nuclear Medicine, Medical Center—University of Freiburg, 79106 Freiburg, Germany; 9Department of Radiation Oncology, Kliniken Maria Hilf, 41063 Mönchengladbach, Germany; Andreas.Kuesters@Med360Grad.de; 10Department of Nuclear Medicine, Helios University Hospital Wuppertal, 42283 Wuppertal, Germany; marco.tosch@helios-kliniken.de; 11Department of Medicine, Faculty of Health, University of Witten/Herdecke, 58448 Witten, Germany; 12Department of Radiation Oncology, Marienhospital, 70199 Stuttgart, Germany; Thomas.Hehr@vinzenz.de; 13Department of Nuclear Medicine, Marienhospital, 70199 Stuttgart, Germany; Susanne.Eschmann@vinzenz.de; 14Department of Radiation Oncology, Klinikum Mutterhaus der Boromäerinnen, 54290 Trier, Germany; YvesPierre.Bultel@mutterhaus.de; 15Department of Radiation Oncology, University Hospital Magdeburg, 39120 Magdeburg, Germany; peter.hass@med.ovgu.de; 16Department of Radiation Oncology, Charité—Universitätsmedizin Berlin, 13353 Berlin, Germany; alexander-henry.thieme@charite.de; 17Department of Radiation Oncology, University Hospital Mainz, 55131 Mainz, Germany; Marcus.Stockinger@unimedizin-mainz.de; 18Department of Radiotherapy, Vienna General Hospital, Medical University of Vienna, 1090 Vienna, Austria; Karin.Dieckmann@akhwien.at; 19Department of Nuclear Medicine, University Hospital Mainz, 55131 Mainz, Germany; Matthias.Miederer@unimedizin-mainz.de; 20Department of Nuclear Medicine, Kliniken Schwerin, 19055 Schwerin, Germany; gabriele.holl@web.de; 21Institute of Medical Biostatistics, Epidemiology and Informatics (IMBEI), University Hospital of Mainz, 55131 Mainz, Germany; koenigjo@uni-mainz.de

**Keywords:** non-small cell lung cancer, NSCLC, concomitant chemoradiation, chemoradiotherapy

## Abstract

**Simple Summary:**

Concurrent chemoradiation (cCRT) with a platinum-based doublet, followed by immunotherapy, is the treatment of choice in locally advanced non-small cell lung cancer. A remaining open question is the difference between cisplatin and carboplatin in combination with second and third generation agents for concurrent chemoradiation, as they have a substantially different toxicity profile and data are scarce and inconclusive concerning cCRT. We here present a secondary analysis of the international PET Plan trial in order to assess the efficacy and toxicity of different chemotherapy regimens as well as the difference between the commonly used platinum based agents, cisplatin and carboplatin. All regimens were well tolerated and cisplatin in combination with vinorelbin either as a single dose or daily doses per cycle showed comparable efficacy. Patients treated with carboplatin doublets had a worse survival, but after adjusting for possibly relevant factors, this difference became non-significant, probably due to existing selection bias.

**Abstract:**

(1) Background: The optimal chemotherapy (CHT) regimen for concurrent chemoradiation (cCRT) is not well defined. In this secondary analysis of the international randomized PET-Plan trial, we evaluate the efficacy of different CHT. (2) Methods: Patients with inoperable NSCLC were randomized at a 1:1 ratio regarding the target volume definition and received isotoxically dose-escalated cCRT using cisplatin 80 mg/m^2^ (day 1, 22) and vinorelbin 15 mg/m^2^ (day 1, 8, 22, 29) (P1) or cisplatin 20 mg/m^2^ (day 1–5, 29–33) and vinorelbin 12.5 mg/m^2^ (day 1, 8, 15, 29, 36, 43) (P2) or carboplatin AUC1 (day 1–5, 29–33) and vinorelbin 12.5 mg/m^2^ (day 1, 8, 15, 29, 36, 43) (P3) or other CHT at the treating physician’s discretion. (3) Results: Between 05/2009 and 11/2016, 205 patients were randomized and 172 included in the per-protocol analysis. Patients treated in P1 or P2 had a better overall survival (OS) compared to P3 (*p* = 0.015, *p* = 0.01, respectively). Patients treated with carboplatin had a worse OS compared to cisplatin (HR 1.78, *p* = 0.03), but the difference did not remain significant after adjusting for age, ECOG, cardiac function creatinine and completeness of CHT. (4) Conclusions: Carboplatin doublets show no significant difference compared to cisplatin, after adjusting for possibly relevant factors, probably due to existing selection bias.

## 1. Introduction

Concurrent chemoradiation (cCRT) with a platinum-based doublet is the treatment of choice in locally advanced non-small cell lung cancer (NSCLC) followed by immunotherapy [1]. In the past decade several prospective studies have assessed the role of the chemotherapy sequence as well as the efficacy of different chemotherapy combinations [2,3,4,5,6,7]. As shown in several meta-analyses, concomitant administration of radiotherapy (RT) and chemotherapy (CHT) compared to sequential administration, improves survival, although there is a higher rate of acute toxicities [8,9], whereas a further chemotherapy consolidation did not lead to an improved survival [10,11]. Most comparative studies for concurrent versus sequential CRT used platinum-based doublets such as cisplatin and etoposide or cisplatin and vinca alcaloids, such as vinorelbin, or cisplatin and pemetrexed for non-squamous histology [12]. According to international guidelines (NCCN v3.2020, ESMO), recommended concurrent chemotherapy regimens include carboplatin or cisplatin in combination with pemetrexed, or paclitaxel, or etoposide and or a vinca alcaloid (typically vinorelbine), but there is no consensus regarding which chemotherapy is better, in combination with RT in this curative setting. This is because it has not been investigated broadly [5,7], if some substances might have a better synergism with RT than others. On one hand, although several protocols might be effective for the treatment of NSCLC, there may be differences in the incidence of toxicities in combination with thoracic RT, which may in turn lead to reduced efficacy, when dose reductions are performed to increase tolerance in combination with RT. On the other hand, intensification of both RT and concurrent CHT may also result into excessive toxicity or incomplete treatment [13].

A remaining open question is the difference between cisplatin and carboplatin in combination with second and third generation CHT. Carboplatin, an analog of cisplatin, has a substantially different toxicity profile compared to cisplatin [14,15] and randomized controlled trials and meta-analyses were inconclusive [16,17,18,19,20,21,22,23].

As recently published, the PET-Plan study investigated radiotherapy target volume concepts in the context of cCRT for locally advanced NSCLC [24]. This international multicenter trial showed a possible advantage in loco-regional control for restricted target volumes with no difference in survival or toxicity between treatment groups. The choice of the concomitant chemotherapy was to the discretion of the treating center in order to enable the use of institutional standards. We here present a secondary analysis of this trial in order to assess the efficacy and toxicity of different chemotherapy regimens as well as the difference between the commonly used platinum based agents, cisplatin and carboplatin, in combination with thoracic RT.

## 2. Results

### 2.1. Patient Characteristics

Between 05/2009 and 11/2016, 205 patients were randomized for target volume definition according to the main study question and 172 patients were allocated to the per-protocol (PP) cohort with respect to radiotherapy, which is analyzed here. Median follow up time for the primary endpoint was 29 months. Demographic and clinical details are shown in Table 1.

A total of 31 patients (18%) of the whole PP cohort were treated according to P1, 92 patients (53%) according to P2, 28 patients (16%) according to P3, 1 patient (1%) received no chemotherapy. Twenty patients (12%) received other chemotherapy regimens such as: cisplatin and oral vinorelbine followed with maintenance with oral vinorelbine and cisplatin, analog to the GILT trial [25], carboplatin in combination with paclitaxel or etoposide. One patient received vinorelbine as monotherapy.

Most patients (*n* = 136 patients) received cisplatin-based doublets treated according to P1, P2 and other protocols and; 33 patients received carboplatin-based treatment according to P3 and other protocols. Both groups were relatively well balanced concerning the tumor stage and histology (Table 2), although patients treated with carboplatin had relatively smaller tumors and less weight loss at baseline.

### 2.2. Toxicity

Patients treated according to P1 and P2 developed more grade 3 leucopenia (16% and 15%, respectively), vs. P3 (4%) (Table 3). Only patients treated according to P1 developed grade 4 hematologic toxicities (6% leucopenia, 3% anemia and 3% thrombocytopenia). There were less grade 3 renal toxicities in P1 and P2 (0%) than in P3 (*n* = one, 4%). Patients treated according to P1 and P2 developed grade 3 cardiac toxicities (6% and 2%, respectively) but no grade 4 toxicities, while there were no grade 3 (0%) in P3 except from one patient (4%) with a grade 4 toxicity. Only two patients treated according to P2 (2%) developed a grade 3 pneumonitis vs. 0% in all other CHT regimens.

Overall, comparing cisplatin vs. carboplatin protocols (Table 4), patients treated with cisplatin experienced more grade 3–4 hematologic toxicities. The incidence of grade 3 pneumonitis was slightly higher in the cisplatin cohort (*n* = two patients, 1% vs. 0%). While higher grade 4 cardiac toxicities were seen in the carboplatin cohort (*n* = one, 3% vs. 0%) grade 3 toxicities were more common in the cisplatin cohort (*n* = two, 1% vs. 0%).One patient (3%) treated with carboplatin experienced a grade 3 renal toxicity, while no patients in the cisplatin group developed grade 3 toxicities.

### 2.3. Dose Modifications and Completion of Chemotherapy

The first cycle was administered as planned in 84% of the patients in P1, 80% in P2, 93% in P3 and 81% of the patients in P1, 72% in P2, 82% in P3 received the second cycle as planned. Causes of deviation were mostly renal or hematologic toxicity.

The difference in OS between patients who completed CHT according to protocol (*n* = 121, 70%) vs. those with protocol deviations (*n* = 50, 29%) was not significant (HR 0.70 95% CI: 0.46–1.08, *p* = 0.10). Moreover, there was no significant difference concerning PFS (HR 0.90 95% CI: 0.63–1.32, *p* = 0.58) or LP (HR 0.72 95% CI: 0.40–1.33, *p* = 0.29) between patients who completed CHT and those who did not.

### 2.4. Efficacy

As previously published, until the fixed end of follow up (31 May 2017), 120 patients died, mainly due to tumor progression.

Patients treated with P1 (median OS: 29 months) or P2 (median OS: 37 months), who received vinorelbin and cisplatin, had a significantly better OS compared to P3 (vinorelbin and carboplatin, median OS: 19 months) (*p* = 0.015 and *p* = 0.01, respectively) and there were more long-term survivors in P1 and P2 compared to P3, but no difference could be seen between the different variants of this combination (P1 and P2; *p* = 0.75; Figure 1). Furthermore, there was no difference detected separately between P1 and P2 compared to the other unspecified CHT regimens.

There was no significant difference observed concerning PFS between the different chemotherapy protocols. Patients treated with P1 and P2 had a lower incidence of LP compared to patients treated with protocols other than P1–P3 (*p* = 0.03).

Patients treated with carboplatin (*n* = 33) had a significantly worse OS as opposed to patients treated with cisplatin (*n* = 136) (HR 1.78, 95% CI 1.05–2.89, *p* = 0.033) (Figure 2) and there were more long-term survivors in the cisplatin-group, but no significant difference in loco-regional or distant progression. In order to exclude effects not related to the substance itself, we performed a multivariate analysis, adjusting for age, ECOG, ejection fraction using cardiac ultrasound at the time of randomization, baseline creatinine (cutoff 1.17 mg/dL) and completion of chemotherapy (yes/no), which was exploratively performed. A significant influence of the applied regimen was not seen in the multivariate analysis (Table 5, HR carboplatin vs. cisplatin 1.44, 95% CI 0.76–2.61, *p* = 0.26).

There were no significant differences concerning PFS (HR 1.19, 95% CI 0.74–1.82, *p* = 0.46) or incidence of local progression (HR 1.38, 95% CI 0.62–2.75, *p* = 0.40) between the two platinum-groups.

As the study centers had stated their institutional chemotherapy standards upon study entry and deviation from standard may be seen as surrogate for non-fitness of the patient towards this standard, we exploratively analyzed the impact of keeping this standard or not on survival. For this analysis we included only patients treated in 6 major centers (*n* = 136), of whom 105 patients we treated according to the predefined standard (P1: *n* = 19; P2: *n* = 86), while in 31 patients the treatment deviated from the standard (P3: *n* = 28, other CTH regimens: *n* = 3) which resulted in a worse OS (HR 0.47 of standard vs. non-standard therapy, 95% CI 0.28–0.80, *p* = 0.0065, Figure 3).

## 3. Discussion

Chemotherapy is an integral part in the treatment of locally advanced NSCLC as it improves survival in all subgroups of patients, whether treated with surgery or radiotherapy, as shown in meta-analyses based on individual patient data [27,28]. The three most common chemotherapy regimens used concurrently with thoracic radiation for patients with unresectable stage IIIA and IIIB NSCLC are cisplatin or carboplatin–paclitaxel (PP), cisplatin–etoposide (PE), cisplatin–vinorelbin (PV) and carboplatin with pemetrexed (CP), but currently there is no consensus regarding which chemotherapy is best in combination with radiotherapy in the curative setting. While there are some studies and meta-analyses conducted comparing PE with PP with conflicting results [29,30,31,32] or PE with CP [5], there are no data comparing PV with other cCRT regimens. Moreover, there are conflicting data concerning cisplatin-based compared to carboplatin-based doublets in combination with RT.

In this secondary analysis of the prospective randomized international multicenter PET Plan trial, we evaluated the efficacy of different CHT regimens on outcome and toxicity. There was a significant difference in OS between patients treated with cisplatin–vinorelbin regimens (P1 or P2) vs. carboplatin-vinorelbin regimens. Furthermore, there was a difference in the LRR between patients treated in P1 and P2 vs. other regimens. In order to further clarify these differences, we conducted an analysis comparing patients treated with cisplatin vs. carboplatin-based regimens observing a significant difference in OS after application of those compounds. In multivariate analysis, however, after adjusting for different selection factors such as age ECOG, renal and heart function and the completeness of chemotherapy this difference became insignificant probably due to selection bias, indication that patients without comorbidities received rather cisplatin than carboplatin. This result may contribute to the current discussion. Meta-analyses have shown that carboplatin has comparable efficacy compared to cisplatin in combination with third-generation drugs in the treatment of very advanced disease [17,33]. However, other prospective trials and meta-analyses have shown that cisplatin-based regimens could be slightly more effective in terms of response rates [18,34,35,36,37] or OS [38,39]. Indeed, there is remaining evidence that carboplatin may also not be as effective as cisplatin against micro metastases in lung cancer, as we have learned from the LACE (Lung Adjuvant Cisplatin Evaluation) meta-analysis on adjuvant chemotherapy in stage II and III NSCLC [40,41,42]. In the curative setting where more aggressive therapy may be feasible and warranted, cisplatin might be more effective than carboplatin, but to date there are no prospective data for cCRT. Furthermore, carboplatin dosage, when used in curative treatment approaches in combination with RT, is considerably more difficult than cisplatin dosage [41]. In a Surveillance, Epidemiology and End Results-Medicare registry analysis comparing cisplatin vs. carboplatin-based chemotherapy in elderly patients, carboplatin-based CRT showed similar long-term survival but lower rates of toxicity [43]. Definitive conclusions are hampered by the lack of large-scale trials for direct comparison [42]. Together with the sources cited, our analysis rather supports the view that patient selection and treatment conduction due to comorbidities may be at least as important for overall survival measures as the application of the compound itself. This is underlined by the result of the exploratory analysis of the outcome after standard vs. non-standard treatment with respect to the treating center, where patients receiving non-standard chemotherapy had a significantly worse OS than those receiving the standard CHT regimen. Another important aspect concerning the consolidation with check-point inhibitors [44] is the ability of chemotherapy to elicit immunogenic cell death. Both clinical and pre-clinical data suggest that both cisplatin and carboplatin exert immunostimulatory effects [44,45].

Regarding adverse events, randomized trials and meta-analyses have shown that carboplatin caused more hematotoxicity, thrombocytopenia, and cisplatin caused more nausea/vomiting in stage IV disease [17,33,46]. In a recent Cochrane review [46], there was no statistically significant difference found in renal toxicity (RR 0.52, 95% CI 0.19 to 1.45; I2 = 3%; 3 RCTs; 1272 participants); alopecia (RR 1.11, 95% CI 0.73 to 1.68; I2 = 0%; 2 RCTs; 300 participants); anaemia (RR 1.37, 95% CI 0.79 to 2.38; I2 = 77%; 10 RCTs; 3857 participants); and neutropenia (RR 1.18, 95% CI 0.85 to 1.63; I2 = 94%; 10 RCTs; 3857 participants) between cisplatin-based chemotherapy and carboplatin-based chemotherapy regimens. Two randomized trials performed a health-related quality of life analysis; however, as they used different methods of measurement, we were unable to perform a meta-analysis [46]. One randomized trial reported comparative health-related quality of life data between cisplatin and carboplatin-containing arms, but found no significant differences in global indices of quality of life, including global health status or functional scales [46]. Therefore, in the palliative intent, the choice of platin derivatives would be based upon the toxicity profile, patient’s comorbidities and preferences [46]. In curatively intended concurrent treatments, the toxicity profile might differ due to the synergism with RT. In our analysis, the incidence of grade 4 hematologic toxicities were higher in the cisplatin-based CHT. However, patients treated with carboplatin had more severe cardiac and renal toxicities. This might be explained by the fact that patients with renal or cardiac comorbidities were likely to be selected for carboplatin-based CHT. As to pulmonary effects, only patients treated with cisplatin developed grade 3 pneumonitis. Pulmonary toxicity is an important issue for cCRT, in the case of consolidation with check-point inhibitors, such as durvalumab [44], which are also associated with an increased risk of developing immune-related pulmonary events [44,47]..In meta-analysis of randomized trials [29,31,32,48] comparing PE with PP, there was a higher incidence of pneumonitis in patients treated with carboplatin–paclitaxel, which was not used in our trial, while in a SEER analysis between patients treated with cisplatin vs. carboplatin based protocols, there was no difference in the incidence of pneumonitis [43,44]. Due to the small number of patients and the rarity of this toxicity, it is not possible to draw any general conclusions from our analysis. Another important issue is the toxicity of cCRT profile in elderly patients. Since the incidence of toxicities was lower in patients treated with carboplatin, while showing similar efficacy, it could be considered as an option in elderly patients. Alternatively, sequential chemotherapy schedules could be offered in elderly patients, since they are associated with less toxicity and no statistically significant decrease in efficacy [49]. We could not find any difference in OS, PFS and incidence of local progression between patients treated with cisplatin–vinorelbine protocols according to Vokes et al. [6] with a single dose of 80 mg/m^2^ per cycle, nor according to Semrau et al. [26] with daily doses of 20 mg/m^2^ (5 days per cycle) for cCRT. A direct comparison, concerning the efficacy, with other PV protocols used concurrently with CRT, such as the ESPATÜ trial [50] combining cisplatin 50 mg/m^2^ (day 1, 8) and vinorelbin 20 mg/m^2^ (day 1, 8) is not possible, due to the differences in the RT protocols (once daily vs. twice daily RT), the delivery of induction CHT in the ESPATÜ trial, and tumor resection in a number of the patients. As far as can be judged, due to the induction CHT, patients treated in the ESPATÜ trial may have had a slightly higher incidence of grade 3–4 hematological toxicities as compared to patients treated with P1 or P2 protocols of the PET Plan trial.

PV doublets seem to be well tolerated. However, a direct comparison between PV and other platinum-based protocols for cCRT is hampered, due to differences between prospective trials in this setting, such as total radiotherapy dose delivered, or combination with EGFR inhibitors [51], the use of induction [6,7,50,52] or consolidation chemotherapy [5,52].

Obviously, our analysis has the limitation being a secondary explorative analysis of a trial with another objective. Therefore, we face imbalances between the different sub-groups and cannot rule out more extensive selection effects. Additionally, a quality of life assessment, which might have added more information, was not performed in the trail. However, it is a unique opportunity to investigate the reported issues posed in a large prospective multicenter cohort with quality assured radiotherapy and institutional chemotherapy standards reflecting current guidelines.

## 4. Materials and Methods

The randomized controlled PET-Plan trial (ARO 2009-09) was conducted in 24 centers in Germany, Austria and Switzerland in accordance with the Declaration of Helsinki and was registered in ClinicalTrials.gov, NCT00697333.

Study design, procedures and main outcome results have been published elsewhere [24]. Briefly, patients with histologically proven inoperable locally advanced NSCLC suitable for cCRT were randomly assigned (1:1) to target volume delineation informed by ^18^F-FDG PET and CT (including CT-positive but ^18^F-FDG-negative nodes) plus elective nodal irradiation and tumor-associated atelectasis, if applicable (conventional target group) or target volumes informed by PET alone (^18^F-FDG PET-based target group). The primary goal of the PET-Plan study was to show non-inferiority of ^18^F-FDG PET-based planning compared to conventional target volume planning in cCRT. In this secondary analysis of the PET-Plan trial we focus on the role of different CHT protocols for cCRT.

### 4.1. Patient Selection

The study was approved by the ethics committee of the Albert-Ludwig University Freiburg, (approval date 30.07.2008, approval number: 276/08). Eligible patients were older than 18 years and had histologically or cytological proven inoperable stage II or III NSCLC. All patients had an Eastern Cooperative Oncology Group (ECOG) performance status of less than 3 and an adequate pulmonary, cardiac, renal, and hematological function (as assessed according to local standards) using pulmonary function tests (PFT) and cardiac ultrasonography prior to therapy and were judged to be suitable for cCRT by interdisciplinary consensus.

### 4.2. Radiotherapy

Both treatment groups were planned to receive an isotoxically dose-escalated radiotherapy up to a total dose of 60–74 Gy (2 Gy per fraction), giving highest possible radiotherapy doses while adhering to predefined normal tissue constraints. Photon-RT was given either as intensity-modulated radiation therapy (IMRT) or three-dimensional conformal radiation therapy (3D).

### 4.3. Chemotherapy

Concurrent CHT consisted of a platinum-based doublet, according to actual guidelines at the time of protocol development. A choice between different published protocols was allowed by protocol at the discretion of the treating physician:

ACisplatin 80 mg/m^2^/d (day 1 and 22) and vinorelbin 15 mg/m^2^/d (day 1 + 8 and 22 + 29) according to Vokes et al. [6] (Protocol 1, P1),BCisplatin 20 mg/m^2^/d (day 1–5 and 29–33) and vinorelbin 12.5 mg/m^2^/d (day 1, 8, 15 and 29, 36, 43) according to Semrau et al. [26], (protocol 2, P2)CCarboplatin AUC1 (day 1–5 and 29–33) and vinorelbin 12.5 mg/m^2^/d (day 1, 8, 15 and 29, 36, 43) according to Semrau et al. [26], (protocol 3, P3)DOther platinum-based doublets such as cisplatin, 50 mg/m^2^/d (days 1, 8, 29, and 36) and etoposide, 50 mg/m^2^/d (days 1–5 and 29–33) according to Albain et al. [52], cisplatin 20 mg/m^2^/d and etoposide 50 mg/m^2^/d (days 1 to 5 and days 29 to 33) according to Fournel et al. [8], cisplatin and oral vinorelbine followed with maintenance with oral vinorelbine and cisplatin, analog to the GILT trial [25] or any other platinum-based doublet according to the institutional standard operating procedure of each (others).

Upon study entry, every participating center was asked to define their institutional standard regimen for patients treated within the study.

### 4.4. Response Evaluation and Toxicity

Treatment response and toxicity were assessed during CRT, within one week after the end of radiotherapy, afterwards every 3 months during the 1st year, then every six months up to the 5th year and thereafter annually. Response was evaluated according to predefined criteria and the Response Evaluation Criteria in Solid Tumors (RECIST) by CT and/or ^18^F-FDG PET/CT in case of suspected loco-regional or distant disease progression. Acute toxicity (up to 90 days after start of radiotherapy) was classified according to Common Terminology Criteria (CTC) version 3 and late treatment-related toxicity was assessed according to the Radiation Therapy Oncology Group/European Organisation for Research and Treatment of Cancer (RTOG/EORTC) scoring system.

### 4.5. Statistical Analysis

All randomly assigned patients were included in the intention-to-treat (ITT) set, while the per-protocol (PP) set included only the patients who were treated according to the radiotherapy protocol, having no treatment-relevant protocol deviations in radiotherapy planning and conduction, as identified by the central radiotherapy quality assurance. In order to exclude potential bias from the radiotherapy, we performed the present secondary analysis in the per-protocol set.

For evaluation, the patients were grouped according to the choice of the different chemotherapy protocols by the treating center, according to the use of cisplatin or carboplatin and according to the treatment by predefined institutional standard or not.

To assess and compare the efficacy and toxicity of the different chemotherapy protocols, we analyzed the endpoints as defined for the analysis of the primary study question. The quantitative variables in treatment groups were characterized by median, range, mean and standard deviation (SD). All *p* values are two-sided and are referred as significant if ≤0.05. As this is a secondary analysis of the data, the sample size has not been planned for the analyses presented here. Therefore, the power of detecting effects may not be sufficient for all subgroups. To estimate the uncertainty, we therefore report the 95% confidence intervals in addition to the *p* values. Additionally, as it was a secondary analysis, patients were not randomized according to the different CHT protocols, so adjusting for possible confounders needed to be considered in our analyses. For this purpose, we also conducted multivariate analyses.

Overall survival (OS), progression-free survival (PFS) and incidence of local progression (LP) were estimated by the Kaplan–Meier method. The observations for OS and PFS were censored at date of last contact or end of the study, and for LP at the time of the last imaging. The times are measured starting from randomization. Since we analyze subgroups of the per-protocol group with respect to the radiotherapy, all patients therein live at least until the radiotherapy has been completed. This is not expected to be a source of bias, as the timing of the radiotherapy is equal across all compared groups.

After observing a difference in OS between cisplatin vs. carboplatin, we exploratively applied a multivariable model, adjusting for a number of published factors [33,53], which may affect patient selection for either regime (age, ECOG, ejection fraction at baseline in cardiac ultrasound, baseline creatinine) as well as for completeness of chemotherapy in order to adjust for effects other than those related to the substance itself. Patients receiving cisplatin-based chemotherapy as part of the initial treatment were allocated to the cisplatin subgroup, while patients receiving carboplatin as part of the initial cCRT were allocated to the carboplatin subgroup. The statistical software packages SAS (version 9.4), and R (version 3.5.3) were used for statistical analyses.

## 5. Conclusions

Concomitant chemoradiation for locally advanced NSCLC with cisplatin in combination with vinorelbine either as a single dose of 80 mg/m^2^ per cycle (P1) or daily doses of 20 mg/m^2^ (5 days per cycle, P2) show comparable efficacy, while the second protocol was better tolerated in this patient cohort. Patients treated with carboplatin and patients not treated by predefined institutional standards had a significant worse survival, most probably due to patient-selection and to restrictions in normal tissue function.

## Figures and Tables

**Figure 1 cancers-12-03359-f001:**
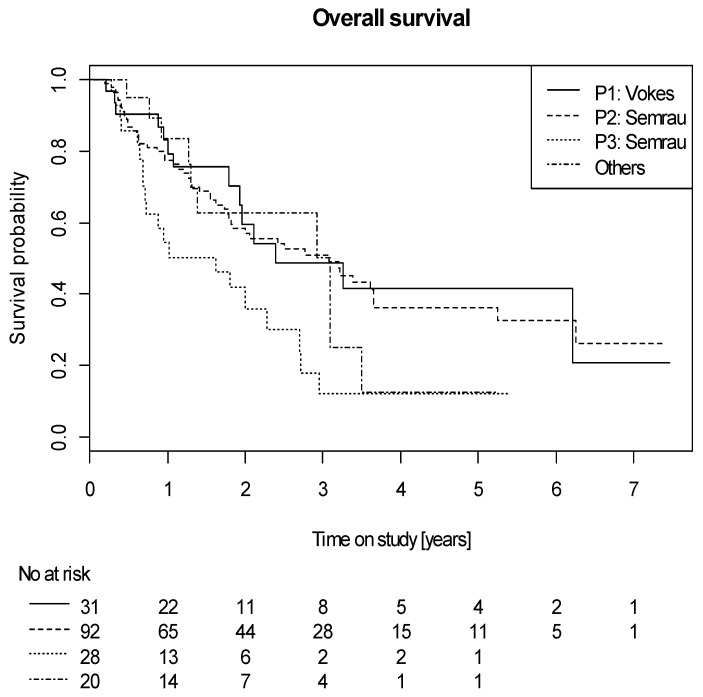
Overall survival according to different chemotherapy protocols.

**Figure 2 cancers-12-03359-f002:**
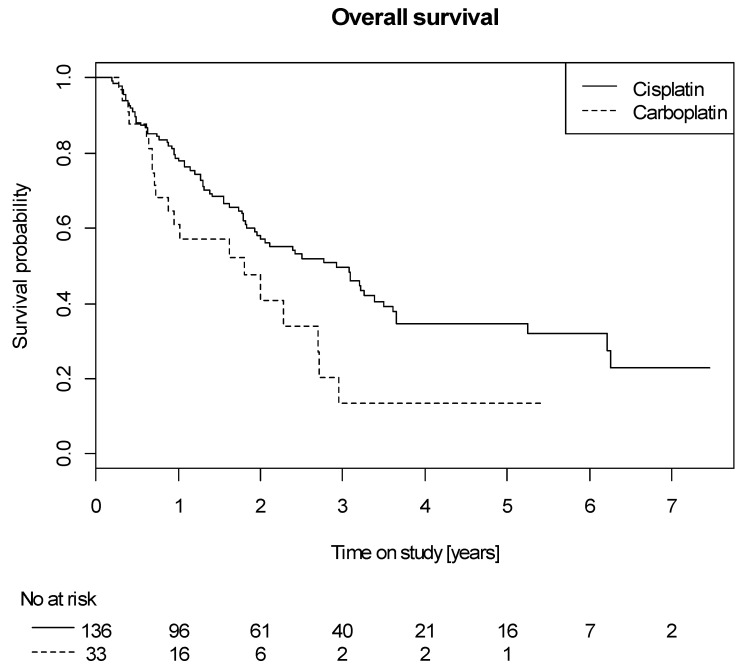
Overall survival of patients with cisplatin vs. carboplatin-based protocols.

**Figure 3 cancers-12-03359-f003:**
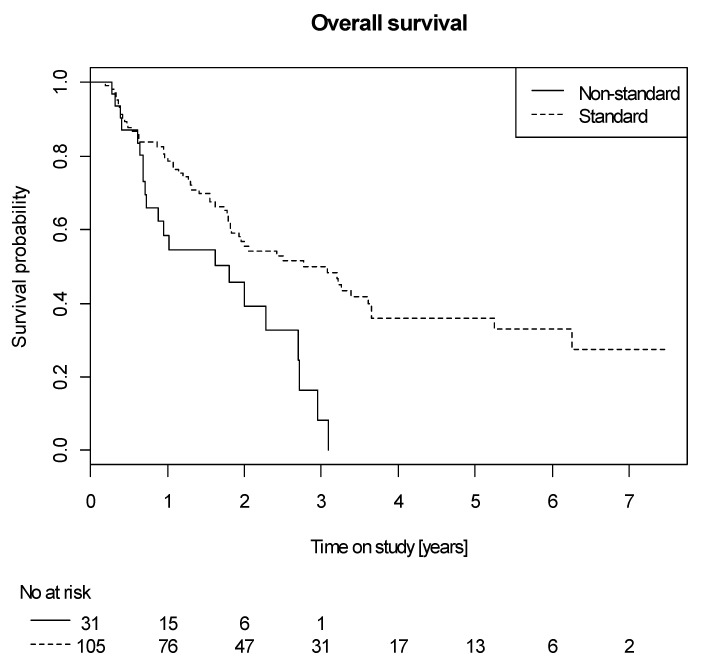
Overall survival of patients treated according to institutional standard vs. non-standard chemotherapy protocols.

**Table 1 cancers-12-03359-t001:** Demographic and clinical details of the patients receiving chemoradiation according to different chemotherapy regimens.

	P1 (*n* = 31)	P2 (*n* = 92)	P3 (*n* = 28)	Others * (*n* = 20)
**Age, years**				
Median (IQR)	61 (57–65)	65(58–72)	70 (63–73)	62 (59–70)
**Sex**				
Male	24 (77%)	68 (74%)	22 (79%)	13 (65%)
Female	7 (23%)	24 (26%)	6 (21%)	7 (35%)
**ECOG performance status**				
0	5 (16%)	12 (13%)	6 (21%)	6 (30%)
1	25 (81%)	70 (76%)	19 (68%)	12 (60%)
2	1 (3%)	10 (11%)	3 (11%)	2 (10%)
**UICC (7th edition) stage at study inclusion**				
IIA	0 (0%)	3 (3%)	2 (7%)	0 (0%)
IIB	6 (19%)	1 (1%)	0 (0%)	1 (5%)
IIIA	9 (29%)	39 (42%)	11 (39%)	3 (15%)
IIIB	16 (52%)	49 (53%)	15 (54%)	16 (80%)
**Histology**				
Squamous cell carcinoma	21 (68%)	49 (53%)	19 (68%)	11 (55%)
Adenocarcinoma	9 (29%)	31 (34%)	6 (21%)	9 (45%)
Large cell carcinoma	0 (0%)	2 (2%)	2 (7%)	0 (0%)
NOS or other subtypes	1 (3%)	9 (10%)	1 (4%)	0 (0%)
Missing	0 (0%)	1 (1%)	0 (0%)	0 (0%)
**GTV primary, mL**				
Median (IQR)	52 (35–113)	73 (38–127)	62 (27–76)	98 (43–237)
**PTV, mL**				
Median (IQR)	428 (326–627)	525 (369–726)	528 (343–695)	682 (452–830)
**Number of PET-positive lymph node stations**				
Median (min–max)	2 (0–7)	3 (0–9)	3.5 (1–7)	3 (0–9)
**Ejection fraction at baseline** **§**				
median (min–max)	60 (50–76)	55 (35–92)	60 (50–79)	60 (45–75)
missing	10 (32%)	4 (4%)	1 (4%)	0 (0%)
**Creatinine in mg/dL at baseline**				
Median (min–max)	0.77 (0.42–1.25)	0.86 (0.5–1.39)	1.1 (0.6–1.77)	0.84 (0.59–1.7)
Missing	0 (0%)	2 (2%)	0 (0%)	0 (0%)
**Weight loss †**				
<5%	18 (58%)	62 (67%)	21 (75%)	13 (65%)
5–10%	2 (7%)	9 (10%)	4 (14%)	1 (5%)
>10%	6 (19%)	16 (17%)	1 (4%)	5 (25%)
Missing	5 (16%)	5 (6%)	2 (7%)	1 (5%)
**Complete administration of chemotherapy**				
Cycle 1	26 (84%)	74 (80%)	26 (93%)	12 (60%)
Cycle 2	25 (81%)	66 (72%)	23 (82%)	10 (50%)

IQR: inter quartile range, ECOG: eastern cooperative oncology group, UICC: union international contre le cancer, GTV: gross tumor volume, NOS: not otherwise specified, Data are reported as *n* (%) or median (min–max) or IQR: inter quartile range, § using cardiac ultrasonography, † 6 months before study inclusion. P1 (protocol 1): cisplatin 80 mg/m^2^/d (day 1 and 22) and vinorelbin 15 mg/m^2^/d (day 1 + 8 and 22 + 29) according to Vokes et al. [6], P2 (protocol 2): cisplatin 20 mg/m^2^/d (day 1–5 and 29–33) and vinorelbin 12.5 mg/m^2^/d (day 1, 8, 15 and 29, 36, 43) according to Semrau et al. [26], P3 (protocol 3): carboplatin AUC1 (day 1–5 and 29–33) and vinorelbin 12.5 mg/m^2^/d (day 1, 8, 15 and 29, 36, 43) according to Semrau et al. [26], * other chemotherapy regimens such as cisplatin and oral vinorelbine followed with maintenance with oral vinorelbine and cisplatin, analog to the GILT trial [25], carboplatin in combination with paclitaxel or etoposide, vinorelbine monotherapy.

**Table 2 cancers-12-03359-t002:** Demographic and clinical details of patients receiving cisplatin vs. carboplatin.

	Cisplatin (*n* = 136)	Carboplatin (*n* = 33)
**Age, years**		
Median (IQR)	64 (58–71)	68 (61–73)
**Sex**		
Male	101 (74%)	25 (76%)
Female	35 (26%)	8 (24%)
**ECOG performance status**		
0	21 (15%)	8 (24%)
1	104 (76%)	21 (64%)
2	11 (8%)	4 (12%)
**UICC (7th edition) stage at study inclusion**		
IIA	3 (2%)	2 (6%)
IIB	8 (6%)	0 (0%)
IIIA	49 (36%)	13 (39%)
IIIB	76 (56%)	18 (55%)
**Histology**		
Squamous cell carcinoma	76 (56%)	22 (67%)
Adenocarcinoma	47 (35%)	8 (24%)
Large cell carcinoma	2 (1%)	2 (6%)
NOS or other subtypes	10 (7%)	1 (3%)
Missing	1 (1%)	0 (0%)
**GTV(Primary) mL**		
Median (IQR)	68.8 (36–127)	66 (28–150)
**PTV, mL**		
Median (IQR)	514 (343–699)	542 (357–737)
**Number of PET-positive lymph node stations**		
Median (min–max)	3 (0–9)	3 (1–7)
**Ejection fraction at baseline §**		
Median (min–max)	60 (35–92)	59 (45–79)
Missing	14 (10%)	1 (3%)
**Creatinine at baseline**		
Median (min–max) mg/dL	0.82(0.42–1.39)	1.03 (0.6–1.77)
Missing	2 (1%)	0 (0%)
**Weight loss in 6 months before inclusion**		
<5%	87 (64%)	26 (79%)
5–10%	12 (9%)	4 (12%)
>10%	26 (19%)	1 (3%)
Missing	11 (8%)	2 (6%)
**Complete administration of chemotherapy**		
Cycle 1	108 (79%)	29 (88%)
Cycle 2	97 (71%)	26 (79%)

IQR: inter quartile range, ECOG: eastern cooperative oncology group, UICC: union international contre le cancer, GTV: gross tumor volume, NOS: not otherwise specified Data are reported as *n* (%) or median (min-max) or IQR: inter quartile range, § in echocardiography.

**Table 3 cancers-12-03359-t003:** Toxicity profiles of different chemotherapy protocols.

	P1 (*n* = 31)	P2 (*n* = 92)	P3 (*n* = 28)	Other * (*n* = 20)
**Leukopenia**				
Grade 0	10 (32%)	22 (24%)	15 (54%)	3 (15%)
Grade 1–2	12 (39%)	53 (58%)	12 (43%)	10 (50%)
Grade 3	5 (16%)	14 (15%)	1 (4%)	5 (25%)
Grade 4	2 (6%)	0 (0%)	0 (0%)	1 (5%)
Missing	2 (6%)	3 (3%)	0 (0%)	1 (5%)
**Anemia**				
Grade 0	17 (55%)	48 (52%)	16 (57%)	8 (40%)
Grade 1–2	11 (35%)	38 (41%)	11 (39%)	10 (50%)
Grade 3	0 (0%)	3 (3%)	1 (4%)	1 (5%)
Grade 4	1 (3%)	0 (0%)	0 (0%)	0 (0%)
Missing	2 (6%)	3 (3%)	0 (0%)	1 (5%)
**Thrombocytopenia**				
Grade 0	23 (74%)	81 (88%)	24 (86%)	14 (70%)
Grade 1–2	3 (10%)	8 (9%)	4 (14%)	5 (25%)
Grade 3	2 (6%)	0 (0%)	0 (0%)	0 (0%)
Grade 4	1 (3%)	0 (0%)	0 (0%)	0 (0%)
Missing	2 (6%)	3 (3%)	0 (0%)	1 (5%)
**Cardiac toxicity**				
Grade 0	24 (77%)	76 (83%)	21 (75%)	19 (95%)
Grade 1–2	0 (0%)	10 (11%)	6 (21%)	0 (0%)
Grade 3	2 (6%)	2 (2%)	0 (0%)	0 (0%)
Grade 4	0 (0%)	0 (0%)	1 (4%)	0 (0%)
Missing	5 (16%)	4 (4%)	0 (0%)	1 (5%)
**Pneumonitis**				
Grade 0	20 (65%)	84 (91%)	25 (89%)	20 (100%)
Grade 1–2	7 (23%)	4 (4%)	3 (11%)	0 (0%)
Grade 3	0 (0%)	2 (2%)	0 (0%)	0 (0%)
Grade 4	0 (0%)	0 (0%)	0 (0%)	0 (0%)
Missing	4 (13%)	2 (2%)	0 (0%)	0 (0%)
**Renal toxicity**				
Grade 0	20 (65%)	54 (59%)	13 (46%)	12 (60%)
Grade 1–2	6 (19%)	35 (38%)	14 (50%)	7 (35%)
Grade 3	0 (0%)	0 (0%)	1 (4%)	0 (0%)
Grade 4	0 (0%)	0 (0%)	0 (0%)	0 (0%)
Missing	5 (16%)	3 (3%)	0 (0%)	1 (5%)

P1 (protocol 1): cisplatin 80 mg/m^2^/d (day 1 and 22) and vinorelbin 15 mg/m^2^/d (day 1 + 8 and 22 + 29) according to Vokes et al. [6], P2 (protocol 2): cisplatin 20 mg/m^2^/d (day 1–5 and 29–33) and vinorelbin 12.5 mg/m^2^/d (day 1, 8, 15 and 29, 36, 43) according to Semrau et al. [26], P3 (protocol 3): carboplatin AUC1 (day 1–5 and 29–33) and vinorelbin 12.5 mg/m^2^/d (day 1, 8, 15 and 29, 36, 43) according to Semrau et al. [26], * other chemotherapy regimens such as cisplatin and oral vinorelbine followed with maintenance with oral vinorelbine and cisplatin, analog to the GILT trial [25], carboplatin in combination with paclitaxel or etoposide, vinorelbine monotherapy.

**Table 4 cancers-12-03359-t004:** Toxicities in patients receiving cisplatin vs. carboplatin.

	Cisplatin (*n* = 136)	Carboplatin (*n* = 33)
**Leukopenia**		
Grade 0	34 (25%)	16 (48%)
Grade 1–2	69 (51%)	16 (48%)
Grade 3	24 (18%)	1 (3%)
Grade 4	3 (2%)	0 (0%)
Missing	6 (4%)	0 (0%)
**Anemia**		
Grade 0	69 (51%)	19 (58%)
Grade 1–2	57 (42%)	12 (36%)
Grade 3	3 (2%)	2 (6%)
Grade 4	1 (1%)	0 (0%)
Missing	6 (4%)	0 (0%)
**Thrombocytopenia**		
Grade 0	112 (82%)	29 (88%)
Grade 1–2	15 (11%)	4 (12%)
Grade 3	2 (1%)	0 (0%)
Grade 4	1 (1%)	0 (0%)
Missing	6 (4%)	0 (0%)
**Cardiac toxicity**		
Grade 0	112 (82%)	26 (79%)
Grade 1–2	10 (7%)	6 (18%)
Grade 3	4 (3%)	0 (0%)
Grade 4	0 (0%)	1 (3%)
Missing	10 (7%)	0 (0%)
**Pneumonitis**		
Grade 0	117 (86%)	30 (91%)
Grade 1–2	11 (8%)	3 (9%)
Grade 3	2 (1%)	0 (0%)
Grade 4	0 (0%)	0 (0%)
Missing	6 (4%)	0 (0%)
**Renal toxicity**		
Grade 0	82 (60%)	16 (48%)
Grade 1–2	45 (33%)	16 (48%)
Grade 3	0 (0%)	1 (3%)
Grade 4	0 (0%)	0 (0%)
Missing	9 (7%)	0 (0%)

**Table 5 cancers-12-03359-t005:** Multivariate analysis of the impact of cisplatin vs. carboplatin together with known selection parameters and completeness of chemotherapy on overall survival.

	Hazard Ratio	95% CI	*p*
**Carboplatin vs. Cisplatin**	1.44	0.76–2.61	0.26
**Age at randomization**	1.007	0.98–1.037	0.63
**ECOG at baseline**			0.94
0	1.00		
1	1.00	0.56–1.92	
2	1.13	0.48–2.58	
**Ejection fraction (%) at baseline** §	0.996	0.973–1.018	0.71
**Chemotherapy complete** (Yes vs. No)	0.72	0.45–1.15	0.17
**Creatinine cut-off 1.17** (≥1.17 vs. <1.17)	1.69	0.84–3.20	0.14

§ in echocardiography.

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
