# Peer review of "Efficacy and Toxicity of Different Chemotherapy Protocols for Concurrent Chemoradiation in Non-Small Cell Lung Cancer—A Secondary Analysis of the PET Plan Trial"

_cancers, 2020, doi:10.3390/cancers12113359_

Round 1
Reviewer 1 Report
Authors show the results on efficacy and toxicity of different chemotherapy
protocols for concurrent chemoradiation in non-small
cell lung cancer.
The paper is interesting but there are some major issues to be addressed
First of all, Statistical analysis section should be implemented. We don't have any information about how authors calculate the study population size to have a statistical significant value. This should be explained clearly.
How did they randomize patients?
Do authors calculate the cumulative dose of platinum therapy? Is there a different in each group?
Is related to RFS or OS?
Is there a QOL analysis? If not should be discussed as limit
Material and methods section should be moved before the results
Is this study a preplanned analysis?
Author Response
First of all, we would like to thank the reviewers for their valuable comments and suggestions.
Reviewer 1:
- First of all, Statistical analysis section should be implemented. We don't have any information about how authors calculate the study population size to have a statistical significant value. This should be explained clearly.
Answer: The statistical analysis section was in the initial version of the manuscript under page 16 line 384 to page 17 line 407, now page 19-20. The title of the statistical analysis section was in the last line of page 16, while the rest was on the next page so it was not easy for the reviewers to see. We apologize for that. We have now corrected that problem. As it was a secondary analysis of the trial there was no additional calculation for the study population concerning the chemotherapy protocols. The calculation for the PET Plan trial is reported in the initial manuscript Nestle U, et al. Lancet Oncol. 2020 as stated under materials and methods (372-373), which is referenced in the methods section. Additionally, we state now more clearly that it is a secondary analysis and we discuss the implications of this there(line 438-444).
- How did they randomize patients?
Answer: We did not perform a randomization concerning the chemotherapy protocol. This is now also mentioned explicitly in the materials and methods section. (line 377-379).For evaluation, the patients were grouped according to the choice of the different chemotherapy protocols by the treating center, according to the use of cisplatin or carboplatin and according to the treatment by predefined institutional standard or not. The randomization concerning the target volume definition is described under materials and methods (lines 372-377).
- Do authors calculate the cumulative dose of platinum therapy? Is there a different in each group?
Answer: Thank you for the suggestion, this an interesting point that we have indeed considered.
We did not perform such calculation at the end as there was no difference in OS, or PFS between patients treated with cisplatin protocols (P1 and P2, page 10 Efficacy). The only difference in OS was between cisplatin and carboplatin but a comparison between cisplatin and carboplatin doses is not possible. We did also perform a statistical analysis between patients receiving chemotherapy as planned vs those that did not receive chemotherapy as planned and did not find any difference in OS, PFS or LP (s. 2.3 Dose modifications).So we did not feel that this calculation would add some additional information.
- Is related to PFS or OS?
Answer: As stated above there was no difference in OS, PFS and LP in patients treated in P1 and P2.
- Is there a QOL analysis? If not should be discussed as limit
Answer: Thank you for this comment. This is an important point and we have addressed this in the discussion under line 322-326 and limitations of the study (lines 363-364).
- Material and methods section should be moved before the results
Answer: Unfortunately, we can’t move the section before the results because according to the template of “Cancers” the material and methods section should be placed after the discussion.
- Is this study a pre-planned analysis?
Answer: This was not a pre-planned analysis. We have made more clear now that this is a secondary analysis in the methods section (line 338-444).
Reviewer 2 Report
The manuscript reports on a secondary analysis of PET Plan trial [Nestle U, et al. Lancet Oncol. 2020] and assessed the efficacy and toxicity of different chemotherapy protocols for concurrent chemoradiation in locally advanced NSCLC.
The interest on the existence of differences between the various chemo-radiotherapy schedules is growing due to the recent approval of durvalumab as consolidation therapy in non-progressing patients after chemo-radiation. Therefore, the identification of schedules associated with less toxic effects, especially pneumonitis, are eagerly awaited.
Minor comments:
- Discussing the role of concurrent chemo-radiotherapy in elderly, the results of the recent analysis of ALLIANCE clinical trials should be mentioned [Maggiore RJ, et al. J Geriatr Oncol. 2020]
- Radiotherapy-induced pneumonitis during concurrent chemo-radiotherapy could hamper the use of consolidation therapies, such as durvalumab, which are associated also with the risk of developing immune-related pulmonary events [Naidoo J, et al. Clin Lung Cancer 2020]. It would be of interest to discuss this topic and highlight the differences in pulmonary toxicity between the different chemo-radiotherapy protocols.
Author Response
First of all, we would like to thank the reviewers for their valuable comments and suggestions.
1) Discussing the role of concurrent chemo-radiotherapy in elderly, the results of the recent analysis of ALLIANCE clinical trials should be mentioned [Maggiore RJ, et al. J Geriatr Oncol. 2020]
Answer: Thank for this suggestion. We have addressed this important issue in our discussion. Lines 342-346.
2) Radiotherapy-induced pneumonitis during concurrent chemo-radiotherapy could hamper the use of consolidation therapies, such as durvalumab, which are associated also with the risk of developing immune-related pulmonary events [Naidoo J, et al. Clin Lung Cancer 2020]. It would be of interest to discuss this topic and highlight the differences in pulmonary toxicity between the different chemo-radiotherapy protocols.
Answer: Thank you for this important point, we have discussed it now more thoroughly in our discussion. Lines 334-337.
Reviewer 3 Report
Authors should:
- better clarify chemotherapy protocols in each arm
- critically analyze the outcomes explaining the significance by referencing published research
- revise all figures provided, as Kaplan-Meier curves seem quite uncommon in their left upper section
- improve punctuation and language\style (check the attached .pdf file)
- consider and discuss recent published studies that show similar trends

Author Response
First of all, we would like to thank the reviewers for their valuable comments and suggestions.
Authors should:
- better clarify chemotherapy protocols in each arm.
Answer: All chemotherapy protocols are listed under the section “chemotherapy” under materials and methods (lines 396-411). We did not perform a randomisation concerning the chemotherapy. We have stated this more clearly in the statistical analysis section. lines 434-440
- critically analyze the outcomes explaining the significance by referencing published research
Answer: We have revised the discussion accordingly. For example lines 310-313, 317-327, 335-337, 343-346.
3) revise all figures provided, as Kaplan-Meier curves seem quite uncommon in their left upper section
Answer: It may be confusing that the line of the KM curve stays stable in the beginning at the 1.0. This is because we measure the time from randomization and we analyse only patients which have completed radiotherapy. Therefore, all patients live until the end of the radiotherapy. We explain this now in the section of materials and methods. We also revised the Kaplan-Meier curves and changed the label of the x axis in the Kaplan-Meier curves to “Time on study” to make this clearer, which we explain more clearly also in the statistical analysis (lines 447-451).
4) improve punctuation and language\style (check the attached .pdf file)
Answer: Thank you very much. We have revised the manuscript improving punctuation and language/style (lines 113,115, 136, 137, 140, 17-77, 179, 181-182,210-211, 224,227,240, 241, 273-274,278, 281, 283-285,291,300-304,337, 341,347, 426-431.
5) consider and discuss recent published studies that show similar trends
Answer: We have revised the discussion accordingly. For example lines 310-313, 317-327, 335-337, 343-346.
Round 2
Reviewer 3 Report
Thank you for the efforts you made.
I believe the manuscript has been significantly improved as all the changes requested have been made.